# Development Work in Healthcare: What Supportive and Deterrent Factors Do Employees Working in a Hospital Department Experience in an Improved Work Environment?

**DOI:** 10.3390/ijerph18168394

**Published:** 2021-08-08

**Authors:** Susanna Perä, Therese Hellman, Fredrik Molin, Magnus Svartengren

**Affiliations:** 1Region Västmanland, 721 89 Västerås, Sweden; susanna.pera@regionvastmanland.se; 2Department of Medical Sciences, Occupational and Environmental Medicine, Uppsala University, 752 37 Uppsala, Sweden; fredrik.molin@ipf.se (F.M.); magnus.svartengren@medsci.uu.se (M.S.); 3IPF, The Institute for Organizational and Leadership Development, Uppsala University, Bredgränd 18, 753 20 Uppsala, Sweden

**Keywords:** organizations, hospital departments, workplace, health services, work environment

## Abstract

Work-related mental health issues, accounting for high worker absenteeism in the world’s developed economies, are increasing, with the main cause being workplace conditions. The health services sector is especially experiencing great problems with this, because of challenging psychosocial working conditions. The aim of this study was to explore employees’ experiences of development work with a focus on the work environment within a hospital department with an outspoken special development assignment. The special assignment was decided by the highest management at the hospital and concerned work environment, caring processes, and ways of organizing the work. Eleven employees completed two individually semi-structured interviews, approximately 7 and 13 months after the start of the special assignment at the department. Interviews were analyzed using thematic analysis. The results reveal that both internal and external aspects influence the development work and highlight the importance of viewing the local development work in relation to how the rest of the organization functions. Important factors and conditions for a supportive and change-friendly work culture are discussed, as well as the need to plan for integration and change to create conditions for successful implementation of the results from organizational development and change initiatives.

## 1. Introduction

People’s work environment has long been known as an important factor for health. Work-related mental illness is increasing in the world’s developed economies, which also accounts for an increasing share of sick leave [1]. A stressful psychosocial work environment has been shown to have significant negative consequences for employees’ health [1,2]. Correlations between work in stressful psychosocial work environments and health problems have been shown for stress-related illnesses [3], fatigue [4,5], depression [6], general mental illness [7], sleep problems [8], cardiovascular disease [9], high blood pressure [10], stroke [11], musculoskeletal problems [12], and sick leave [13]. In addition to the personal consequences of ill health for employees, it also entails large costs for employers, in the form of both ill-health-related expenses and direct production losses due to psychosocial work environment factors [14].

Previous research has shown both risk and health factors in a workplace psychosocial work environment [2,15,16,17,18,19]. Even though previous research has focused on important supporting factors for work environments [20,21,22,23,24], it has proven to be difficult to implement these factors in real work situations. In other words, there are knowledge gaps between theory and practice in how to implement initiatives for better work environments in workplaces [22,25]. One reason for the implementation difficulties is that work environment problems are not remedied quickly and that organizations tend to fail with long-term implementations [26]. Another challenge is that it is usually difficult to find suitable interventions for a specific workplace as the interventions tend to work differently in different contexts [27,28]. Studies have also shown difficulties in prioritizing work environment initiatives, even when the changes could save time and increase productivity [29].

The health sector is reported to be one of the most vulnerable sectors for work-related stress [30]. A national inspection effort showed that heavy workload, stress, and stressful patient relationships create ill health and sick leave in healthcare [31]. Previous research has paid attention to nurses working in hospital settings, and high levels of work-related stress and oxidative stress were found [32], as well as emotional exhaustion, depersonalization, and decreased personal performance [33]. According to Kovner et al. [34], 17.5% of nurses tend to leave their jobs within the first year. Nurses who have chosen to resign from their jobs have explained it on the basis of psychosocial abuses, in the form of inhumane working conditions that have had negative health effects, the betrayal and rejection of leaders, the effects of work on private life, and the diminishing and ostracizing behavior of colleagues [35]. There seems to be a need for initiatives to strengthen employees’ trust, wellbeing, and work environment in organizations. Previous research has focused on organizational interventions aiming to prevent work-related stress and psychosocial unhealth. An after-action review is one example of an organizational intervention focusing on daily reflections in groups, in which special situations of various character that have occurred during a work shift are brought up and discussed. This intervention has been shown to prevent work-related stress [36]. However, in general, research on organizational interventions has shown varying results and limited evidence regarding how work-related stress might be effectively prevented [37,38]. Thus, the knowledge on effective interventions in healthcare is still limited [39]. Furthermore, the effectiveness of an intervention also depends on how it is designed, implemented and adapted to the specific context in which it will be used [40]. The challenges of implementing sustainable changes in the healthcare sector are well known [41] and might be a consequence of, for example, low organizational maturity for change [42], poor intervention fit [43], and lack of involvement by employees and their immediate managers [44]. These factors are known to be crucial in efforts and interventions aimed at improving the work environment [23]. Since healthcare is a complex business in an ever-changing environment, work environment initiatives adapted to this are needed [21].

In an attempt to achieve better work environment in a hospital setting, a hospital in Sweden started a project in which one of their departments was given a special assignment; the department has a regular care assignment, as well as a special assignment, for the development and improvement of the work environment. This study, thus, explores a project approach with a high foundation in the everyday tasks of healthcare and in which the employees are highly involved. All development work was initiated and run by the hospital itself.

The aim of this study was to explore employees’ experiences of development work with a focus on the work environment in a hospital department having a special development assignment: What supportive and deterrent factors do employees working in a hospital department experience while working for an improved work environment?

## 2. Materials and Methods

In order to investigate employees’ experiences of development work in a hospital department with a special assignment for improved work environment, a qualitative study was carried out. In the original design of the study, three interviews were planned in order to grasp the participants’ experiences of change over time while working in the project. However, due to the Covid-19 pandemic that began in early 2020, the third interview were not feasible to conduct. The employees at the department were relocated, and their work task was shifted. Due to these circumstances, it was not feasible to study change over time. Two interviews were still conducted, which give insight into the participants’ experiences of development work during the first year of the project. The project was approved by the Regional Ethics Committee in Uppsala, Sweden (Project reference number: 2019-00948). Written informed consent was obtained from all employees.

### 2.1. The Study Context: A Hospital Department with a Special Developmental Assignment

The study was conducted at a hospital in one of Sweden’s major cities. A major challenge that this hospital is facing relates to the shortage of nurses, which has been identified as the hospital’s main reason for closed hospital beds. Furthermore, the hospital has identified a need to deliberately improve the work environment, and their future goals are, for example, to become a prosperous organization through improved participation, increased employee engagement, and strengthened skills provision. Based on their challenges and visions, the highest management at the hospital decided on a 3 year project that concerned development and change of work environment, caring processes, technical solutions, and ways of organizing the work. A surgical department was assigned as the pilot department for the project. According to the project plan, the purpose of the current initiative was to create an attractive workplace where new ideas and working methods should be tested, implemented, and disseminated, both within their own area of operations and within the hospital’s other operations. The idea was that employees from all professions, together in the department, would collaborate and identify what changes need to be made regarding the business’s routines, processes, and patient flows. This implied that the surgical department was also assigned fewer hospital beds than ordinary departments in order to have time for the development work. Furthermore, they had a larger mandate to organize their work in other ways than was usually done in the hospital. However, as time passed, it was not fully possible to keep the lower number of beds at the department due to the high patient pressure at the hospital.

### 2.2. Participants 

All staff in the department were informed of the study and invited to participate. Full participation in the study required employees to take part in two interviews. Eighteen employees at the department accepted participation from the start. This is a clear majority of employees, although the exact number of employees in the department at the time of inclusion is unknown. One employee dropped out from the study before the first interview as the interviewer did not get in touch with the person. A further six people later withdrew from the study due to reasons such as changing jobs, high levels of stress, and time constraints.

Eleven employees completed the study, comprising 10 women and one man. All employees worked either as specialist nurses (*n* = 3), nurses (*n* = 6), or assistant nurses (*n* = 2). The average age of the employees was 34.2 years (range: 24–51 years). The employees had worked an average of 5.5 months in the department at the time of the first interview (range: 0.5–7 months) and had an average of 6.1 years in their professions (range: 3 months–26 years). 

### 2.3. Data Collection

All employees were interviewed on two occasions. The first interview was conducted 7 months after the start of the special assignment for the department and the second interview 6 months later (i.e., just over 1 year after the start-up).

The interviews were conducted according to semi-structured interview guides (see Table 1). The first interview was made up of open-ended questions, focusing on the employee’s general experiences of working in the department, experiences of different psychosocial work environment factors, and what they would like to change in the workplace in the future. The second interview used the same semi-structured interview guide with general questions as a base. In addition, they were also partly individually designed on the basis of previous interviews and responses taken from a diary that the participants answered every second week. The diaries had a similar focus as the interviews but enabled a deeper understanding of the ongoing process that could be followed up in the second interview. In addition to the general interview guide, the second interview, thus, used more personalized questions addressing the individual experiences of the interviewee.

The interview guide was piloted in the first interview and thereafter discussed in the research group in order to consider whether any changes were needed to clarify questions. However, only minor changes were needed, and the first interview was, thus, included in the study material.

All interviews were conducted by experienced researchers within the field of medicine, occupational therapy, psychology, and organizational behavior. The interviews lasted between 25 and 50 min. They were recorded and transcribed verbatim afterward. 

### 2.4. Data Analysis

The data gathered from the 22 interviews were analyzed thematically, using Braun and Clarke’s five steps for thematic analyses [45]. The analysis were primarily conducted by the first author (S.P.) with tight supervision from T.H. and F.M. during all steps. In the later phases of the analysis, M.S. was involved with asking critical questions regarding the analysis and emerging themes. The analysis began with a basic reading of the interview transcripts. In the next step, all interviews were encoded sentence by sentence into summary codes. The codes were then compared with each other for each separate employee and categorized into summary categories. Once all interview codes were categorized, the employees’ categories and their underlying codes were compared to each other in order to build larger categories based on the aim of the study. Finally, when processing and sorting the larger categories, the five themes of the study emerged. During the work process with the thematic analysis phases, the authors repeatedly went back and forth across the encoded, categorized, and thematized material, as well as the basic transcriptions, to ensure that the results were based on the employees’ statements and not influenced by the authors’ understanding or other assumptions. Throughout the process, notes were also kept on the reflections made on the material in order to capture valuable insights along the way.

In the analysis and the results report, Shenton’s criteria for credibility in qualitative studies [46] were used to guide the study. In addition to the quality assurance of the analysis, the work process was described in order for it to be replicated. Disagreements that emerged in the interviews regarding the themes of the study were highlighted, and context details were added so that readers can better understand how similar this study’s context is to other contexts with which one might want to compare the results.

The emergent themes and results were communicated to the respondents and presented at a staff meeting, ensuring respondent validation (also referred to as a member check) [47]. The use of a diary between the first and the second interviews also allowed for validation of the interview responses. Triangulation was further achieved using different methods of data collection (interviews and diaries) and the different scholarly disciplines represented in the research team [48].

In the analysis, the following themes emerged: “a shared identity of being development-focused nurtured a creative work environment”, “employee-driven change management facilitated the development”, “difficulties in bringing the development work into focus”, “the high workload hampered the work”, and “difficulties in involving people outside of the department in the development work”.

## 3. Results

From the thematic content analysis, five themes emerged that constituted several subcategories (see Table 2).

### 3.1. Theme 1: A Shared Identity of Being Development-Focused Nurtured a Creative Work Environment 

The employees described a common development identity as a supporting factor in the development work. Many employees contrasted this with the resistance they experienced from development initiatives in previous workplaces.

*“If someone comes and says that now we’re going to try this, it’s easier to get people to join in than it was in the last department”. (…) “There was a frustration where I worked earlier in that I could get a job from a manager, I could make a decision, the manager could approve the decision, but then the employees didn’t do that anyway”. (…) “Formal power is the same here, but it’s easier to get changes through here because there are more people in the staff group who are willing to try other things”*.(P1)

#### 3.1.1. Common Incentives

Many of the employees have actively applied to the department to work with development. The development focus has not escaped anyone, and employees express a clear incentive to want to contribute to creating a better and more sustainable work environment for healthcare professionals. Some justify their choice of workplace by the fact that they enjoy development work, while others applied to the department due to their dissatisfaction with the current conditions at the hospital. The hospital’s problems with high staff turnover based on poor work environment were cited by several as the reason why the department has been created.

*“There is a problem with traditional care: especially with staffing and making people feel comfortable, so that they feel they want to stay. You want to be able to staff the hospital with your own staff, and we have been tasked with testing new ways of working and new technology”*.(P1)

The fact that the manager’s recruitment has focused on finding people who want to work with development work also seems to have strengthened the feeling that those who work in the department have common incentives which make them suitable for the task. For some, the anchoring and the importance of the targets also seem to have been further strengthened in that the department is spared from the burden of care that other departments have from the start.

*“Our work task is to develop working methods. We, thus, started with quite few and easy patients in order to focus on our work task. At this point, we are just opened on weekends because in the beginning we were closed”*.(P5)

When the burden of care in the ward later increases, the feeling of being able to work in accordance with the common incentives decreases (see Theme 3: difficulties in bringing the development work into focus).

#### 3.1.2. A Supportive and Developmental Work Culture

A positive and developmental work culture between colleagues was described as a major source of wellbeing and continued development focus even under strained conditions. The open discussion and decision-making climate, the focus on what they themselves can influence, and the willingness to change and adapt to current circumstances were highlighted as important factors.

*“I think it’s a question of attitude: that you are prepared for new things to come, to try new things, and that it changes a little bit in everything from time to time”. (…) “There may be things that need to be solved there and then, but it is then possible to develop further by changing routines or working methods to address the problems”. (…) “We follow up on how it works after a while. Either it works okay or it doesn’t work and then we change a little bit and try for a few more weeks”*.(P8)

The employees also highlighted the helpful and supportive work culture in the working group as an important reason why they enjoy working at the department. They appreciated the openness in being able to ask questions, as well as the fact that there is a sense of security in being able to get both practical and emotional support when needed. Something that was highlighted by several as unique in the department’s group climate is that problems and potential conflicts are solved directly. The working group’s positive attitude toward diversity and different perspectives also emerged as a strength for development. 

*“I think that we have quite high ceilings and that we raise problems directly if you experience any problem within the staff group. In case of irritation, we confront each other and ask what this is all about. We have a pretty open discussion about most things with a light-hearted atmosphere, I would say”. (…) “It’s okay to have different opinions and ideas and to express them”*.(P9)

Even under increased workload in the department, most employees maintain a continued strong experience of a supportive and developmental work culture. However, some believe that there is not enough time to give and request support to the extent they would have liked due to the strained burden of care in the ward.

#### 3.1.3. A Change-Promoting Work Environment 

Another important factor for the common development identity is the change-promoting work environment based on the practical and organizational conditions that are present in the department in order to be able to work with development. This includes, for example, that the department has a manageable workload for care work, as well as sufficient staffing, and that there is time set aside for development work. Furthermore, a good division of labor is highlighted on the basis of structured tasks and a clear division of labor between the professions.

*“We divide the patients into like half/half, so the nurses help each other over their patients. If I have like finished mine but the other have not, then I try to help her and so the assistant nurses will help each other with their chores. In other departments, if the nurses are done with their tasks, they help their assistant nurse; you can do that too, but in the first place you should try to help those in the same profession then instead of taking another profession”*.(P3)

Having a development support manager who also focuses on developing the leadership was seen as a change-promoting aspect for the working group. As the burden of care in the department increases, the experience of a change-promoting work environment, as well as the feeling of a shared identity of being development-focused for several employees, decreases (see Theme 3: Difficulties in bringing the development work into focus).

### 3.2. Theme 2: Employee-Driven Change Management Facilitated the Development

#### 3.2.1. The Entire Working Group Is Involved in the Development Work

All employees in the department are described in some way as being involved in direct development work. Employees also experience a high opportunity for participation through the open social climate in the problem and improvement discussions that are ongoing in the workplace in everyday life. The possibility of open decision making between the manager and employee was also highlighted as participatory.

*“I still think we have high expectations here. If anyone comes up with something that’s feasible, it’s pretty quick to get it through”*.(P2)

Several development initiatives were ongoing at the same time. Here, joint planning and follow-up days seem to have had a gathering effect. Although one is not directly involved in certain development parts, the days provided a good overview of to whom one can turn to share their views on the various improvements that are being developed. However, some did not feel that the planning days were comprehensible and expressed that they lack an overview and sense of opportunity for participation in the various things that happen. 

*“When you sit down and discuss together, you get everyone’s opinions gathered together so that we can work in the same direction. You can see that something is happening with what we think, so I think it has been good that we have had those days”. (…) “Without those days, it probably wouldn’t be the case that everyone got to participate in the same way; then, maybe only some people would have taken the initiative and done things while others had fallen away”*.(P4)

Some also experienced the regular workplace meetings as good forums for participation, while others experienced them as being too time-constrained and too controlled by managerial information being relayed to the employees.

#### 3.2.2. The Change Work Is Based on Employees’ Insights and Initiatives

Employees feel able to control development work on the basis of the needs, problems, and insights they encounter in their workday. Many development ideas seem to start in everyday problems and workplace discussions between colleagues before they are raised to managerial discussions and then decided upon directly or at the regular workplace meetings. Other ideas come via a digital idea box that both staff and patients can write to. The employees testify to having a high sense of room to maneuver and decide mandates when it comes to development issues that they can control in the department. The focus is mostly on the improvements that can be made within the department, and there is an attempt to bring things outside of their control higher up in the organization. In addition to spontaneous improvement discussions and the idea box, daily reflections at the end of the shifts (around patient safety, workload, and team communication) seem to have played a major role in the employee-driven development work.

*“During the reflection, you can bring up what has worked well and why and what has worked poorly and why. We write things down and then (the manager) reads it and then he can see patterns in what doesn’t work well and what works well and bring those things up for discussion at APT (the regular workplace meetings)”*.(P3)

*“I think it’s good to let go of a little bit more of your aggression or frustration from a bad workday when you’ve had to show that it was a bad day and write why”. (…) “I feel good then that we still had time to reflect on what we could have done better and what we can do better in the future, so that it doesn’t happen like this again”*.(P9)

The new technical solutions tested in the department come mainly from higher up in the organization but still seem to be well received by the working group. Technology testing is seen as part of the assignment, and the working group expressed feeling quite free in designing how technology is used and they also often influence how it is designed and programmed.

#### 3.2.3. Method Development Built into the Working Day from the Beginning

Another supporting factor for the employee-driven change work is that the method-developing routines were built into the working day from the beginning. Examples include getting documented group reflections at the end of the work shifts from the beginning. It was also made clear before the start-up that the divisions of labor between the different professions would be reworked in order to optimize the use of everyone’s skills.

*“We’ll help each other out. We understand each other and have worked a lot with work shifting and doing the right things”. (...) “Everyone understands everyone’s value. It’s not that hierarchical; you really look at each other’s skills and everyone’s involved”*.(P6)

Similarly, it seems to have been clear that the department would build new work routines from the start. Here, the employees’ varying backgrounds were also seen as a strength as they had nothing in common to fall back on.

*“It is a completely different spirit [in this department] in terms of development and improvement in work than it is in ordinary cases. // Everyone is open minded to development and change work; you are not stuck in the same track, focusing on why we cannot do as we have done before; here, the focus is more on development”*.(P8)

### 3.3. Theme 3: Difficulties in Bringing the Development Work into Focus 

#### 3.3.1. Difficulties in Influencing External Organizational Decisions

Employees felt that they have very little or no opportunity to influence organizational decisions that hamper the development and are made outside of the department. Several described a lack of control in relation to the external factors that affect their development missions. Examples of such factors are decisions about overcrowding in the ward, the need to receive patient groups for which they are not qualified, and the two departmental moves that they were required to do at short notice between the two interview sessions.

*“We have been at three physical locations. Even though, one location was just over the summer, it has taken quite a lot of energy from us as a working group, I think. And it has also meant that we had to, there are a lot of things that have sort of stopped, you can say that a lot of the energy is spent on moving and get settled; you have to get in place; you have to get back to your way of working”*.(P10)

Several employees described difficulties in not having enough time and resources to make the regular care work in itself. The workload from the care work is considered by many to be too high. The obvious primary focus for the department is to try to cope with the care work at the expense of the development mission. The problem of greater burden of care and priority development work increased between the two interview sessions. 

*“It’s a bit difficult to have two assignments, to kind of care for patients and do care development at the same time. Because somehow patient work has to go first”*.(P1)

The task force also felt frustrated that their manager is similarly forced to prioritize the organization’s demands over the needs of the department. The employees had a view that the manager’s ambition from the beginning was to be a present manager who actively supports the development, but they felt that, since the start of the development department, they were instead engrossed with other things. The manager was perceived by many as absent and stressed. This was described as having consequences for the department’s work processes, regarding both the care work and the development work.

#### 3.3.2. To Start Something New Demands Much Time and Resources

In addition to increased demands in the form of increased pressure to provide care, the difficulty of starting from scratch was also highlighted, in connection with both the department’s start-up and the two department relocations in which the department was involved. In the first interview, the employees expressed an understanding for the demanding start-up of the department, as they saw a meaning and a purpose with the opening. However, when employees were later forced to restart again after two demanding relocations, it became more difficult to find new motivation. It was described as very time-consuming and resource-intensive to build new routines and to find new roles and collaborations in relation to their new surroundings. Many employees described having to devote a lot of working time to tasks that do not belong to their profession in order to make the care day work. This involves, for example, having to try to compensate for the lack of knowledge that exists in the working group on new patient groups and dealing with a lack of knowledge about ordering procedures in order to make daily care needs and deliveries work. At the start of the department, the numerous introductions of new colleagues were also mentioned as a stressful and time-consuming factor to continue in parallel with the department’s routines, trying to build it up from scratch. Employees felt a burden of being in an unstructured and unclear work situation.

*“It has taken quite a toll on us as a working group, and there are many things that have stopped because a lot of the energy is spent on moving”. (…) “In concrete terms, this means that you cannot continue so much with the development work”. (…) “The focus just becomes to survive and put out fires instead of making it better”*.(P10)

### 3.4. Theme 4: The High Workload Hampered Their Work

#### 3.4.1. Developmental Factors due to Work Overload 

The overload felt by employees from care work and other external factors was reported to have emotional, cognitive, and other health-impacting consequences that hamper the development at the department. Several employees described feelings of stress, feelings of inadequacy, and frustration at not being able to get a sustainable work situation as the worst things about their work. Some felt emotionally affected even when they returned home and needed to use their private network and/or turn to their colleagues in their spare time to deal with this. There was also frustration that the development work that could help them out of their health and safety problems could not be prioritized in the long term, in favor of solving urgent problems here and now. In addition, the consequences of work overload made it more difficult to do good development work even if time is freed up. Some employees described themselves as being visibly affected by the high workload. Employees reported having memory problems, difficulty sleeping, and stress and pain problems as a result of the work situation. The pace of work and decision-making ability were noted as having deteriorated with higher loads and fewer recovery opportunities. Several members of the task force also expressed frustration with the impact of late scheduling on privacy, as well as the lack of recovery that scheduling also provides. Some employees were considering leaving the department due to privacy reasons and health effects, even though they were comfortable with the tasks and atmosphere in the workplace.

*“I try to think in 6 month periods to cope: I’ll try another 6 months and we’ll see. Because I think that work is so fun and rewarding and evolving and it’s fun to go to work, but then I can come home and be exhausted”. (…) “I still think it’s my job that affects me that I have these problems; I sleep much worse as well, as you’ve probably seen. Yes, but it’s the kind of stuff that makes me start to wonder if it’s worth it even though I think it’s so fun. It obviously wears on my body too, not just on the social”*.(P9)

The experiences of work overload, its consequences, and its impact of scheduling on private life were not shared by everyone in the working group. At the time of the second interview, some people also felt that they were okay in the department and that they were still more spared from work overload than the rest of the hospital.

Another development barrier due to work overload was the perceived shortcomings of the manager’s leadership. The manager was not perceived to be able to keep up with their assignments, and there were also concerns about its impact on both their leadership conditions and health conditions. Some experienced support and good dialogue opportunities with their manager, while others found it difficult to get a hold of the manager and felt that they were not listened to when they raised problems.

*“I experience that, when you talk to the manager, it is usually the case that it flows very much off or that it is turned and twisted; I kind of do not really think that any action is taken when you raise some issues”*.(P11)

Lack of foresight in planning and communication deficiencies, both around planning and unresolved problems, were also highlighted by some employees as consequences of the manager’s high workload. When an assistant manager was appointed, however, some decision-making processes and dialogue opportunities were perceived as having improved.

#### 3.4.2. Unmet Expectations

Other employees were considering changing jobs on the basis of the high workload, combined with not being allowed to work with the development work to the extent they expected. There was a growing sense that they are increasingly becoming a regular ward over time, and many felt less and less able to identify themselves as working in a department with a special assignment. Disappointment and unmet expectations emerged in the working group according to the contradiction between how employees thought that the work arrangement would be when they were recruited to and started working in the department, compared to how it has become. Some were concerned about the department’s survival. Others remained more hopeful about the future of the department. They believed that they still managed to carry out some development work and believed that the development focus would be able to increase in the future.

*“Now we’re a little more of a throwaway department”. (…) “We have to take a lot more patients and we have to take a lot more patients who are not our category. Over there… (in the first department before the moves) we were very spared, and it felt more like we were a project”*.(P7)

#### 3.4.3. Difficulties in Getting More Complex Development Work in a Strained Care Workday

Several employees expressed that there is too little development and that they lack the development focus. Others felt frustrated that the development focuses on the wrong things. Technological development continues, while, for some, the more important development of working methods had to take a back seat. Similarly, there was frustration that simple and quick solutions are squeezed into everyday life, while more holistic changes are not being prioritized. Some employees also reflected that it would be much easier to identify problems that they need to solve than to pay attention to success factors.

### 3.5. Theme 5: Difficulties in Involving People outside of the Department in the Development Work

#### 3.5.1. Difficulties in Involving Doctors in the Processes of Change

The nonparticipation of doctors in the working group and in the development work came to light as a difficulty in several interviews. The role of doctors is an important part of the care work and is affected by much of the ongoing development work. At the same time, the doctors are organized in another part of the business outside of the department. There are different doctors doing the rounds and patient work every week. The doctors do not participate in the daily reflections about the work that has been performed during the workday, even though that was the plan from the beginning. The employees in the department had different pictures of why this was so. Some perceived doctors as positive to change and complicit to the extent they could be. Organizational difficulties and lack of time were then seen as the main difficulties for doctors in being able to participate in development. Others felt that the doctors themselves want to be outside the working group and that they as a profession are skeptical about changes, with hesitancy in wanting to help do things differently when the working group proposed it. Perceived hierarchies between doctors and other healthcare professionals were also addressed as a possible explanation.

*“I don’t think we would have really understood what a problem it is not to have a department doctor. We have different doctors every week and to then innovate the rounds and do something completely different than what is usual is quite difficult when there are different people every week who are going to be involved in this”*.(P1)

#### 3.5.2. Difficulties in Cooperation with Neighboring Departments 

Some of the work conditions and development tasks depend on cooperation with the department’s care neighbors. For example, common premises for kitchens, washrooms, medicine rooms, and storage rooms are always shared with a neighboring department. The employees described difficulties in building new routines and collaborations with their care neighbors in several different ways. Staff turnover and the large proportion of hired staff in neighboring departments hamper the ability to make joint method development efforts. Territorial thinking between departments was seen as another problem.

*“We had to move down and work in another ward, and it was very tricky because it was not our kind of patients and then it was not our routines and then we did not find in the premises either. They did not have our material that we needed for our patients; there was a lot of running and picking up and borrowing from other departments”*.(P5)

However, this seemed to apply mostly during the daytime, and the collaboration was perceived to work better during the night shifts. Some employees also pointed out that neighboring departments with similar patient groups would probably facilitate collaboration and understanding of each other’s situations.

#### 3.5.3. Lack of Understanding and Interest from Others in the Hospital Organization 

The interviews also revealed experiences of skepticism and a lack of understanding from others in the hospital organization as to why the department with a special assignment exists. Some employees believed that those in the surroundings questioned the existence and benefits of the department. The employees felt that the perceived skepticism was mainly due to a lack of understanding and knowledge about the background for the department’s special assignment and its goal to ultimately help everyone in the hospital. The overcrowding in the department, without taking into consideration that the department’s care capacity should not be as high as that of other departments, could also be linked to this. 

*“I think that the idea is that we will try to find better ways of working and get a sustainable workplace, yes working methods that you can spread further so that we can have functioning units. And then it’s for everyone’s sake, but I don’t think everyone understands that’s the case”*.(P2)

There were also employees who noticed positive curiosity from people in different positions outside the department.

Several employees described expectations that information dissemination would take place, and they had a view that it is in their development mission to spread knowledge and help with the implementation of successful development results in other departments. In practice, however, this has not happened, except for a few instances based on specifically requested topics such as scheduling. Some employees linked the lack of information to the fact that they have not been given sufficient conditions to work with development. There were concerns about being met with resistance and that the improvements that the department develops would not be used by other departments. Some believed that other departments’ working methods are deeply rooted and that improvement proposals from their department would probably not be appreciated. In particular, there were concerns linked to trying to spread improvements in their own surgical field, as the comparison between the different working methods of the departments and work cultures would be clearer than in departments with other patient groups.

*“But I think it may be that; Why do you think you are better than us?” (…) “It would certainly also be difficult even if we were a medical department to reach out to the rest of the medicine division”*.(P2)

At the second interview, the employees also did not perceive that there was a dissemination plan; rather, they felt that they stopped talking about knowledge dissemination and the purpose of implementation in the department.

## 4. Discussion

This study qualitatively explored employees’ experiences of development work with a focus on the work environment within a hospital department. In the analysis, the following themes emerged: “a shared identity of being development-focused nurtured a creative work environment”, “employee-driven change management facilitated the development”, “difficulties in bringing the development work into focus”, “the high workload hampered the work”, and “difficulties in involving people outside of the department in the development work”.

These five themes confirm much of the previous research on both success factors and deterrent factors in work environment change. The importance of participation, social support, and employee-driven change work has been emphasized by many researchers [16,23,24,49,50], as well as the difficulties in getting the development work in focus when the ordinary business needs to be prioritized [29,51]. The developmental consequences of work overload are also already known [52,53], as well as the significance of perceived organizational support for employee engagement [54] and the significance of recurring reflections for an improved work environment [36,55,56].

What has been less researched in the past are the supportive and deterrent factors based on the specific conditions that arise from this special initiative of highlighting the development work through a department with a special developmental assignment. Examples of important factors are the common development identity created in the group, as well as the contrast and difficulties that manifest themselves in relation to how the rest of the organization works. The fact that the organization’s regular operations interfere with development work, with health and developmental consequences as a result, also says something about the difficulties that need to be addressed in order to succeed with work environment improvements in hospital contexts. The hospital environment is ever-changing, and work environment initiatives adapted to this are needed. Overall, the factors that the employees and managers in the department themselves have been able to control seem to have become success factors for the development work. What is common for the deterrent factors is that they are outside the department’s own governance and often involve difficulties in relation to regular hospital operations.

Achieving a common development identity in a working group is not something to be taken for granted. The feeling of being a cohesive work group is often based on factors that are stabilizing, which often lead to clashing with a group’s propensity for change [57]. The fact that the working group in this study was clearly selected on the basis of its interest in development and built its group identity on being a change-prone group has probably had an impact on the working climate and development work created.

In addition to the creation of a common identity in being a department with a special assignment, the working group also described skills in communicating, agreeing, and harnessing the group’s collective resources. These skills are crucial for a positive group climate and for a group to be good at problem-solving [57,58]. The developmental culture that exists in the working group in the department is not something that a regular health department can count on from the beginning when implementing changes. Employee and first-line manager resistance to change and fatigue after previous unsuccessful attempts at change can be an obstacle when attempting new work environment improvements [49].

An important factor for overcoming resistance to change has been shown to be employee participation in the ongoing change. Participation has been shown to have a positive impact on employees’ attitudes toward change when previous experiences are valued [59] and continuous feedback is given on how the ongoing change work is progressing [49]. The work environment efforts in the department itself seem to have been well integrated, and it has become natural for the employees to reflect on in their work day. Integration is often linked to high employee participation in work environment efforts and to the fact that the focus of work environment management is on learning to recognize and deal with real and current problems in the business [60]—two areas that the study’s department has succeeded in doing.

The remainder of the organization’s understanding and participation in development work does not seem to have been in focus from the employees’ perspective, and the department’s dissemination of knowledge to ordinary activities has not become a natural part of the development work during the first year. The reason that work environment efforts need to be broadly anchored and performed in harmony with the entire organization is that each change is followed by chain reactions that affect the entire organization, according to Conner, as cited by Al-Haddad and Kotnour [61]. Furthermore, one needs to realize that, in the specific workplace, there are limited opportunities to influence the root cause of work environment problems, as causes and solutions to psychosocial work environment problems can be found elsewhere in the organization [62]. This was demonstrated in this study by the difficulties that employees experienced in influencing the external organizational decisions, which hindered development work. It has previously been shown that organizations’ conditions for change and methods of change need to work together, and that change and implementation planning are two important aspects to take into account in order for the desired effect to be achieved [61,63].

According to the employees’ experiences of other departments’ lack of understanding, a clearer communication of the department’s purpose and a clearly communicated implementation process, involving individuals in receiving departments in the change work, would probably have facilitated a successful implementation of the department’s work.

### Limitations

When applying the results of this study, it is important to be aware that the employees’ experiences in this study were captured in the first year of the department’s special assignment. Unfortunately, the planned third data collection point was not feasible to conduct due to the rising Covid-19 pandemic. This study, thus, offers insights into the supporting and deterrent factors of the initial year in the development work; however, it cannot give a picture of the supporting and deterrent factors of the entire project time. Within the group, processes at the department may go through new phases during the project, and the interaction with the rest of the organization may change and be affected in different ways during the remaining 2 years. The study also explores a specific setting within a hospital context and, thus, results cannot be generalized beyond this setting.

Another factor to take into account when interpreting the results is that this study focused on the employee perspective. Whether management teams or managers higher up in the organization have done anchoring work, implementation plans, or worked according to specific working methods that employees did not perceive was unclear. The purpose of this study, on the other hand, was to capture the employee perspective. Although implementation plans and anchoring work have been carried out by others in the organization, the results revealed that the employees in the hospital department did not perceive or notice their effects.

Another possible limitation of the study is the small number of participants and that not all of the department’s employees participated in the study. In total, seven employees chose to discontinue the study. Whether these individuals had different approaches and experiences of the development work is not known. As with all research involving human subjects, the voluntary nature of the study may give a biased representation of the studied phenomenon. Nevertheless, as the findings in this qualitative study were drawn from a concrete level of interview data to a more conceptual level, we believe that our findings could be valuable for the purpose of contributing new insights and practical considerations when aiming to reinforce the work environment and work processes in hospital settings. The two different interview sessions conducted with each employee further strengthened the results of the study as they probably increased the employees’ conditions in terms of feeling safe and being understood in the interview situation and the subject, whereby the interviewers also had more opportunities to capture key experiences.

## 5. Conclusions

The five themes emerging from the analysis confirm previous research on both success factors and hindering factors in the work environment change. On the other hand, the common development identity created in the group, as well as the contrast and difficulties that manifest themselves in relation to how the rest of the organization works, provides new insights into what specific supporting and deterrent factors may arise from conducting change work through a department with a special developmental assignment in a hospital context. This study underlines the importance of employee participation and group skills in communicating, agreeing, and harnessing the group’s combined resources in order to create a favorable development climate. The study also draws attention to the need for integration and change planning in order for the results of the development work to have the conditions for a successful implementation.

For further research, it is desirable to follow more departments with a similar developmental assignment to get more information over time regarding what similarities and differences may arise from different contexts and conditions. Long-term follow-ups throughout the project are also recommended to be able to share the experiences and insights of working groups through all phases of the projects. Furthermore, a future research focus on knowledge dissemination and implementation of the department’s work environment improvements could make an important contribution to understanding how such a department’s development work can be effectively integrated and implemented into ordinary operations.

## Figures and Tables

**Table 1 ijerph-18-08394-t001:** General questions in the semi-structured interview guides.

Interview	Questions
First interview—general questions	How would you describe working in your workplace? What do you think about how the workplace is organized? What do you think about how you treat each other? Would you like to change something in the workplace? Is there anything else you would like to tell us that we have not asked about?
Second interview—general questions	What characterizes your workplace today in general? Describe how your workplace is organized? What do you think about how treat each other? Would you like to change something in the workplace? Do you think you have time to perform your “special tasks”? Have you set aside time to perform your “special tasks”? Do you have time for reflection? What do you think about the fact that you have a developmental assignment? Is there anything else you would like to tell me that I have not asked about?
Second interview—individual follow-ups from diary and first interview	You mentioned this… What has happened regarding this since the last interview? How did you think when you answered…? Can you develop/clarify…?

**Table 2 ijerph-18-08394-t002:** Themes and subcategories of supportive and deterrent factors experienced by employees in the department in the work for an improved work environment.

Themes	Supporting/Deterrent	Subcategories
A shared identity of being development-focused nurtured a creative work environment	Supporting factor	Common incentives A supportive and developmental work culture A change-promoting work environment
Employee-driven change management facilitated the development	Supporting factor	The entire working group is involved in the development work The change work is based on employees’ insights and initiatives Method development built into the working day from the beginning
Difficulties in bringing the development work into focus	Deterrent factor	Difficulties in influencing external organizational decisions To start something new demands much time and resources
The high workload hampered their work	Deterrent factor	Developmental factors due to work overload Unmet expectations Difficulties in getting more complex development work in a strained care workday
Difficulties in involving people outside of the department in the development work	Deterrent factor	Difficulties in involving doctors in the processes of change Difficulties in cooperation with neighboring departments Lack of understanding and interest from others in the hospital organization

## Data Availability

The data presented in this study are available on request from the corresponding author. The data are not publicly available due to the data consisting of transcribed interviews which could not be completely anonymized.

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
