# Peer review of "Development Work in Healthcare: What Supportive and Deterrent Factors Do Employees Working in a Hospital Department Experience in an Improved Work Environment?"

_ijerph, 2021, doi:10.3390/ijerph18168394_

Round 1

Reviewer 1 Report

The paper investigates a very relevant topic today that also very interest issues. But, this article should be refine the writing and specification on this research process. 

The data collected from 22 interviewers whose are the participant interview that should be detail list out and did/no match your project planning it.

This paper was focus on qualitative method where no find out from the context of manuscript how to do triangulation test what is basic for reliability and validity analysis. 

Reviewer 2 Report

I believe that the article is structured and organized with rigor and has the potential to be published. It presents small aspects that should be improved, namely in the abstact, key words and material and methods.

Reviewer 3 Report

Dear authors,

This is a well-written article on a nicely done study. Please add following information:

  • The found aspects can be framed as social capital. It may be worthy considering to add a short section on this in the introduction. 
  • Please add the interview guides. Has there been a pre-test of the guides?
  • In the section "data collection" it is mentioned, that the researchers were experienced. Please add the background of researchers.
  • In the section "data analysis": Please add (in brackets) who was involved in which step.
  • It is possible to additionally interview members of the management team? The manuscript would benefit from the managers' perspective. 
  • Please add a point in the discussion about the voluntary nature of participation.

Kind regards

Round 2

Reviewer 1 Report

  1. This paper investigates a relevant work environment topic today. Specially, the employees working in a hospital is very heavy duty and long time working environment. How to understand the supportive and deterrent factors that is very interest issues.
  2. The new revision was add some related literature for support this article.
  3. The data collected from 22 interviewers whose are the participant interviewer of project that it was point out in the content of new revision. 
  4. This paper was focus on qualitative method. The author add the interview content in Table 1.  

Reviewer 3 Report

Dear authors,

Thank you very much for your revision. Most of my comments were satisfactorily dealt with. 

However, is a pity that additional interviews with the management were not possible. The research question suggests a multi perspective approach in this setting. The managers' perspective would have enhanced the soundness of the whole paper and the interest to the readers of IJERPH.  

I wish you all the best for your manuscript. 

Kind regards

Author Response

This manuscript is a resubmission of an earlier submission. The following is a list of the peer review reports and author responses from that submission.

Round 1

Reviewer 1 Report

  • This is an interesting topic to improve the understanding regarding the health and workplace nexus. However, I have the following reservations and comments:
  • My foremost objection is about sampling and data. The authors did not mention which data sampling technique was used. How questions/checklist was developed and why? What kind of checklist or questionnaire was used?
  • Is it reasonable to extract results from only 11 people and present as a base study?
  • And this number is also gender-biased, i.e., 1 male and 10 females. I am not convinced.
  • Then again why that particular method were used to analyze this very small sort of data?
  • How this population size was determined?
  • How baisness was dealt in data collection?
  • There are many sentences in the abstract which are given under inverted commas. What is the reason behind that? Usually, it is done to avoid plagiarism.
  • Abstract can be further improved considering the ripple effects.
  • Same comment is for conclusions. And also try to convince me that how 11 person’s information can be considered as a base study or conclusive while we are living in the age of “Big Data”?
  • The analysis is too weak.
  • Why not publish it as a Short communication?
  • Keywords should be arranged alphabetically.
  • Only 1 table and no figure at all provided. This does not make a good sense of the science.

Reviewer 2 Report

  1. The authors have used qualitative data information from middle-aged nurses only, as «all employees worked either as specialist nurses (n=3), nurses (n=6), or assistant nurses (n=2). The average age of the employees was 34.2 years (range: 24 years – 51 years)» As a result, I suggest the authors focus the title of their paper on middle-aged nurses by changing the word "employees" (it is quite general) to "middle-aged nurses" (it is more specific and informative) in the title of their paper.
  2. The authors conclude that «the results reveal that both internal and external influence the development work and highlight the importance to view the local development work in relation to how the
    rest of the organization functions.» This sentence is quite general and vague, so I suggest the authors explain it further.
  3. Some sentences in the paper (pages 9 and 10) are very general, e,g., «Several employees express...» (How many? Percentage?), «Several employees describe...»(How many?Two?Three?Five?)»,«Some believe that other departments' working...»(How many? Two?), «Some employees link the lack...» (How many?),... Please, I suggest the authors include numbers or percentages to be more informative and concise.
  4. The world is suffering from the Covid19 pandemic on a global scale, including Sweden, and it strikes me that the pandemic is not even mentioned in the paper. Therefore, I suggest the authors include a future research line on how SARS-CoV-2 influences the employee-driven change in healthcare or another related topic.